# Monocarboxylate Transporter-2 Expression Restricts Tumor Growth in a Murine Model of Lung Cancer: A Multi-Omic Analysis

**DOI:** 10.3390/ijms221910616

**Published:** 2021-09-30

**Authors:** Abdelnaby Khalyfa, Zhuanhong Qiao, Murugesan Raju, Chi-Ren Shyu, Lyndon Coghill, Aaron Ericsson, David Gozal

**Affiliations:** 1Department of Child Health and the Child Health Research Institute, School of Medicine, University of Missouri, Columbia, MO 65201, USA; qiaoz@health.missouri.edu; 2Department of Ophthalmology, School of Medicine, University of Missouri, Mizzou, Columbia, MO 65212, USA; rajum@health.missouri.edu (M.R.); lcoghill@missouri.edu (L.C.); 3Institute for Data Science and Informatics, Department of Electrical Engineering and Computer Science, University of Missouri, Columbia, MO 64110, USA; shyuc@missouri.edu; 4Department of Veterinary Pathobiology and Metagenomics Core, University of Missouri, Columbia, MO 65212, USA; ericssona@missouri.edu

**Keywords:** MCT2, metabolome, microbiota, GC−MS, LCMS, tumor, lactate, RNA-seq, RNA

## Abstract

Monocarboxylate transporter 2 (MCT2) is a major high-affinity pyruvate transporter encoded by the SLC16A7 gene, and is associated with glucose metabolism and cancer. Changes in the gut microbiota and host immune system are associated with many diseases, including cancer. Using conditionally expressed MCT2 in mice and the TC1 lung carcinoma model, we examined the effects of MCT2 on lung cancer tumor growth and local invasion, while also evaluating potential effects on fecal microbiome, plasma metabolome, and bulk RNA-sequencing of tumor macrophages. Conditional MCT2 mice were generated in our laboratory using MCT2^loxP^ mouse intercrossed with mCre-Tg mouse to generate MCT2^loxP/loxP^; Cre^+^ mouse (MCT2 KO). Male MCT2 KO mice (8 weeks old) were treated with tamoxifen (0.18 mg/g BW) KO or vehicle (CO), and then injected with mouse lung carcinoma TC1 cells (10 × 10^5^/mouse) in the left flank. Body weight, tumor size and weight, and local tumor invasion were assessed. Fecal DNA samples were extracted using PowerFecal kits and bacterial 16S rRNA amplicons were also performed. Fecal and plasma samples were used for GC−MS Polar, as well as non-targeted UHPLC-MS/MS, and tumor-associated macrophages (TAMs) were subjected to bulk RNAseq. Tamoxifen-treated MCT2 KO mice showed significantly higher tumor weight and size, as well as evidence of local invasion beyond the capsule compared with the controls. PCoA and hierarchical clustering analyses of the fecal and plasma metabolomics, as well as microbiota, revealed a distinct separation between the two groups. KO TAMs showed distinct metabolic pathways including the Acetyl-coA metabolic process, activation of immune response, b-cell activation and differentiation, cAMP-mediated signaling, glucose and glutamate processes, and T-cell differentiation and response to oxidative stress. Multi-Omic approaches reveal a substantial role for MCT2 in the host response to TC1 lung carcinoma that may involve alterations in the gut and systemic metabolome, along with TAM-related metabolic pathway. These findings provide initial opportunities for potential delineation of oncometabolic immunomodulatory therapeutic approaches.

## 1. Introduction

Monocarboxylate transporters (MCTs) are members of the solute carrier (SLC) family (SLC16) of proteins, comprising 14 isoforms. MCTs are expressed in many different tissues, and are involved in the regulation of fundamental cellular processes, such as glycolysis and fatty acid homeostasis, as well as other key metabolic pathways [1,2,3]. Of the 14 isoforms identified, proton-dependent MCTs 1–4 have been extensively studied due to their importance in transporting L-lactate, pyruvate, and short-chain fatty acids in a wide variety of tissues [4]. Relevant to the current study, many of these MCT isoforms are upregulated in tumor tissues, making them attractive targets and biomarkers for many different types of cancer [1,2,3,5]. In a recent study in lung cancer patients, a higher expression of MCT2 was associated with increased cell senescence, suggesting that increased MCT2 activity may favorably affect the clinical course of the disease [6]. These findings are congruent with the assumption that the presence of a hyperglycolytic phenotype, which requires higher lactate, and pyruvate utilization may be associated with higher overall survival [7,8]. Furthermore, the expression of MCT2 in tumors has been linked to its ability to regulate glutamine-derived TCA-cycle flux, particularly of α-ketoglutarate, and can suppress mitochondrial respiration and decrease ATP production to generate a more susceptible tumor environment [9]. However, the opposite findings have also been reported in experiments involving prostate cancer. Indeed, Valenca et al. [10] showed that a heightened expression of MCT2, particularly when co-localizing with peroxisomes, fostered cellular proliferation of prostate cancer cells. In human cancers, MCT2 are strongly expressed in the cytoplasm of colorectal cancer cells, indicating a possible role within intracellular organelles, such as the mitochondria [11]. In addition, MCT2 is the primary isoform expressed in human glioblastoma multiform and glioma-derived cell lines [12]. In glycolytic tumors, MCTs promote the efflux of lactic acid, as important players in the maintenance of the tumor intracellular pH, avoiding the routing to apoptosis, and providing the favorable microenvironment conditions for invasion [13,14]. Thus, the role of MCTs in general, and more particularly of MCT2, in the tumor properties in vivo remains unclear.

The gut microbiome can modulate the host immune system both locally and systemically [15], and can regulate many functions of the tumor-bearing meta-organism, often through immunomodulation [16]. In healthy adults, most of the endogenous bacteria are represented by two phyla, *Firmicutes* and *Bacteroidetes*, which account for approximately 90% of the resident microbiota [17]. Several studies have shown that changes in the gut and oral microbiota may contribute to carcinogenesis in mouse models and human patients [18,19]. Other studies have shown that specific bacteria present in the intestinal tract can have systemic effects on the immune system by activating pro-inflammatory or anti-inflammatory pathways, which could alter the risk of cancer in multiple organs [20]. Depletion of the gut microbiome in pancreatic cancer and melanoma models, and manipulation of the gut microbiome in other cancer systems modified the tumor burden [21,22,23]. Similarly, cancer growth is promoted by a proinflammatory environment created by the gut microbiota [24].

Gut microorganisms and their metabolites may migrate to other parts of the body via the circulatory system, causing an imbalance in the physiological status of the host, and the secretion of various neuroactive molecules through the gut−brain axis, gut−hepatic axis, and gut−lung axis to affect inflammation and tumorigenesis in specific organs. The immunosuppressive status of the tumor microenvironment remains poorly defined due to a lack of understanding regarding the function of tumor-associated macrophages (TAMs) [25]. TAMs are crucially implicated in tumor progression and metastasis, and favor tumor cells to modify the microenvironment and promote tumor growth, angiogenesis, invasion, and metastasis, as well as suppress the antitumor immune response [26]. A multi-omics approach revealed distinct tumor immune microenvironment contributing to immunotherapy in lung adenocarcinoma [27]. The mechanisms by which gut microbiome interacts with the immune system and affects cancer progression are unclear. In this context, the emerging role of MCT2 in cancer energetics and the parallel metabolic role played by changes in the gut microbiota let us to hypothesize that systemic knock down of MCT2 (KO) in a mouse model of syngeneic lung cancer would not only alter the tumor characteristics, but may also induce significant differences in the gut microbiome and in the fecal and plasma metabolomes. Using bulk RNA sequencing (RNA-seq), we also explored changes in the gene expression of TAMs associated with manipulation of the host MCT2 expression.

## 2. Results

### 2.1. MCT2 Immunoreactivity

To ascertain that the MCT2 protein is down-regulated in the presence of tamoxifen, we conducted immunoblotting in several tissues, including the testis, cortex, and visceral adipose tissues. Compared with vehicle-treated mice, tamoxifen-treated mice showed significant reductions in the MCT2 protein expression in the testis *p* < 0.001, visceral fat *p* < 0.01, and cortex *p* < 0.03, respectively (Appendix A).

### 2.2. Tumor Growth and Local Invasiveness

MCT2 KO mice exhibited an enhanced tumor weight (1646.38 ± 390.41 mg) compared to CO mice (injected with saline) after 24 days (1141.44 ± 540.71 mg; *p* = 0.006, Figure 1). In addition, tumor volumes were significantly increased in KO mice (3882.06 ± 1313.67 mm^3^) compared to the CO animals (2298.31 ± 1094.59 mm^3^; *n* = 16/group, *p*= 0.006; Figure 1), along with increased local invasiveness (i.e., disruption of the tumor capsule; 14/16 in KO and 6/16 in WT; *p* < 0.01), suggesting that a reduced MCT2 expression promoted increased tumor growth and regional invasiveness.

### 2.3. Electron Microscope

Since mitochondria are implicated in the process of tumor biology, which includes alterations of cellular metabolism and cell death pathways, fresh tumor tissues obtained from KO and CO mice were minced, fixed, and imaged using electron microscopy (Figure 2). Mitochondrial changes associated with more of an electron−lucent mitochondrial matrix, swollen mitochondria, and disrupted cristae were apparent in tumors from KO mice compared to CO. Mitochondrial swelling with partial or total cristolysis suggests that the ability of neoplastic cells to generate ATP by mitochondrial oxidative phosphorylation may be diminished.

### 2.4. RNA-seq in Tumor-Associated Macrophages

Next, we purified the TAMs from tumors harvested in KO and CO mice, and performed bulk RNA-seq. The heatmap of the top 100 genes (50-up-regulated, red color, and 50-down-regulated, blue colors) are shown in Figure 3. Gene Ontology (Table 1) and KEGG pathways (Table 2) for those genes. Metabolic pathways that were differentially expressed following Gene Set Enrichment (GSE) analysis included the Acetyl-coA metabolic process, activation of immune response, B-cell activation and differentiation, cAMP-mediated signaling, glucose and glutamate processes, T-cell differentiation, and response to oxidative stress.

### 2.5. Fecal Microbiota

Fecal microbiota composition profiles were analyzed using the 16S rRNA sequencing-based method in samples from both groups. Bacterial community patterns and principal coordinate analysis (PCoA) were compared, and 460 amplicon sequence variants (ASVs) were identified. The ASVs achieving significance in both KO and WT are listed in Appendix A. We also included the phylum, class, order, family, and genus for each ASV (Appendix A). PCoA revealed marked differences in the fecal bacterial composition in KO and CO groups based on Bray−Curtis similarities, PERMANOVA (*p* = 0.016; Figure 4A).

Significant differences in the *α* diversity of the genus-level gut microbiome evaluated by the richness index (Chao 1 index, *p* = 0.04) and bacterial diversity index using Shannon (*p* = 0.02) emerged (Figure 4B). The dominant phyla were Firmicutes (40%), Bacteroidetes (14%), Actinobacteria (2%), Proteobacteria (1%), Proteobacteria (1%), and Verrucomicrobia (1%) (Appendix A and Figure 4C). Fecal microbiome analysis of KO mice showed that the *Firmicutes*/*Bacteroidetes* ratio was six times higher compared to CO. The relative significant phylum is shown in Figure 4C. Firmicutes and Bacteroidetes showed significant differences with increased Firmicutes and decreased Bacteroidetes levels in KO mice (Figure 4C). At the phylum level, the predominant bacterial taxa in the feces of both groups were *Firmicutes and Bacteroidetes*, and the most abundant families were Ruminococcaceae, Muribaculaceae, and Lachnospiraceae. Hierarchical clustering of samples was based on the relative abundance of the 50 ASVs based on the lowest *p*-values following ANOVA of all ASVs comparing KO and CO (Figure 4D).

### 2.6. Fecal and Plasma Metabolomes

To identify the fecal and plasma metabolome features, untargeted metabolome profiles were generated on KO and CO samples using UPLC-MS. For the fecal metabolome, PCA based on the UPLC-MS analysis of the fecal metabolome resulted in the detection of 1738 compounds on Bray−Curtis similarities, PERMANOVA, on KO and CO (*p* = 0.0009; Figure 5A). The ANOVA of these compounds detected differences in the abundance of 129 compounds (Appendix A). The Volcano plot highlighted the significant metabolites in KO vs. CO with a fold change of >1.5 (Figure 5B). The top six metabolites yielding the lowest *p*-values based on *t*-test in KO compared to CO were either up-regulated (C5H11NO2, C24H41NO4, C17H43N7O6S2, and C24H36O3) or down-regulated (C27H27N3O7 and C4H7NO2; see Figure 5C).

The plasma metabolome was also acquired by UPLC-MS and 1858 compounds were identified by retention time and mass/charge (*m*/*z*). No significant differences based on Bray−Curtis similarities, PERMANOVA (*p* = 0.56), were found in KO compared to CO mice (Figure 6A). The data in Figure 6B show the top six serum metabolites (lowest *p* values), all of which withstood correction for multiple tests.

### 2.7. Comparison between Fecal and Plasma Metabolomes

Data from untargeted UPLC-MS revealed 794 unique metabolites in the plasma, 1024 unique ASVs in the feces, and 113 were in common, as shown in the Venn diagram (Figure 7A). Significant differences were associated with the site (plasma vs. fecal) and treatment (KO vs. CO) among 113 common UPLC-MS features, based on two-way ANOVA; Tukey box plots showing the distribution in each group of the two metabolites with significant treatment-associated differences (Figure 7A,B). The data shown in the heatmap (Figure 7B) summarize the ASVs that were altered in the KO and CO in fecal and plasma samples. Among the 113 common UPLC-MS compounds in the fecal and plasma samples, we identified two metabolites (C4H7NO2 and C5H11NO2) with significant subject-associated differences, as shown in Figure 7C, based on two-way ANOVA and box plots.

### 2.8. GC−MS for Plasma and Fecal Metabolomes

In parallel, GC–MS was utilized to profile the polar and nonpolar volatile fractions of the plasma and fecal samples. In the plasma polar, a total of 74 metabolites were detected in the positive ionization mode in both groups and underwent univariate analysis (Appendix A). There were four differentially expressed metabolites, of which two are known (Myo-Inositol and D-(+)-Glucose) and two have unknown metabolic functions (Appendix A).

For example, Myo-Inositol is involved in inositol phosphate metabolism, while D-(+)-Glucose is involved in starch and sucrose metabolism. However, the data in the Volcano plots show only Myo-Inositol based on the high significant of *p*-value and fold change (Figure 8) and box plot (Figure 8B). For the plasma non-polar, out of 41 metabolites, 3 metabolites were significantly different, of which one has a known metabolic function (Cholesterol) and two are unknown (Appendix A). No significant differences emerged in the Volcano plots.

In the fecal metabolomes, 174 metabolites were identified via GC–MS in the polar fraction, and 22 metabolites showed statistical significance in KO vs. CO (Appendix A), of which there were 11 metabolites with known functions, while the other 11 metabolites had an unknown function (*p*-value <0.05). Volcano plots revealed three significant metabolites, namely Xyluose, Xylose, and unknown, as shown in Figure 9A, and the box plot showed these three metabolites identified via GC–MS in KO and CO (Figure 9B). Hierarchical clustering analysis showed a clear separation of KO and CO mice (Figure 9C).

For the fecal non-polar metabolome, we detected 126 metabolites, of which there were 13 that showed statistically significant differences (Appendix A). Among these metabolites, six were identified with known functions. The known metabolites consisted of pentadecanoic acid, inositol-2-phosphate, 1-hexacosanol, n-octacosanol, and tetradecanoic acid (Appendix A). Volcano plots show eight statistically significant metabolites based on fold changes (2) and *p*-value (log10; Figure 10A). Tukey box plots showed the significant abundance of five metabolites in the controls and KO (Figure 10B). Hierarchical clustering in the fecal polar analysis shows the differentially expressed metabolites based on their retention time and weight mass for the KO and CO fecal samples (Figure 10C).

### 2.9. Metabolic Pathways

The metabolic pathway of the target metabolites identified by GC−MS of both plasma and fecal of KO and CO mice are summarized in the metabolic systems map shown in Appendix A). For plasma polar metabolites, two metabolic pathways were identified, namely inositol phosphate metabolism and starch and sucrose metabolism (Appendix A), while in the fecal polar analyses, 11 metabolic pathways were also identified, including glutathione metabolism, nucleotide sugars metabolism, purine metabolism, transfer of acetyl groups into mitochondria, tryptophan metabolism, and tyrosine metabolism, as well as valine, leucine, and isoleucine degradation (Appendix A).

For the plasma polar metabolites, bile acid biosynthesis was significant (Appendix A), while for fecal non-polar metabolites, six metabolic pathways were identified, including acylcarnitine 15-(3,4-dimethyl-5-pentylfuran-2-yl) pentadecanoylcarnitine, inositol phosphate metabolism, arachidonic acid metabolism, amino sugar metabolism, fatty acid biosynthesis, and amino sugar metabolism (Appendix A). We noticed inositol phosphate metabolism is the common pathway between the plasma and fecal samples. In addition to metabolic pathways, the genes associated with each pathway can be useful for linking multi-omics analyses with other high-throughput technologies. For example, the arachidonic acid metabolism pathway has 61 genes, including *ALOX5*, *LTC4S*, *CYP1A1*, *CYP1B1*, *CYP2C8*, *CYP4F2*, *ALOX5AP*, *GPX1*, *GPX2*, *GPX4*, *ALOX15*, *PTGS2*, *PTGS1*, *CYP1A2*, *CYP2C19*, *CYP2C9*, *CYP4A11*, *DPEP1*, *LTA4H*, *PTGR1*, *DPEP2*, *PTGDS*, *TBXAS1*, and *PTGES3*.

### 2.10. Gene Networks

We constructed gene networks for two metabolites, namely myo-inositol for inositol phosphate metabolism, and propionic acid for tryptophan metabolism (Figure 11). The Regulatory Network of Target Genes was constructed with String software. In this network, several genes are ranked as highly significant in inositol phosphate metabolism, such as *IPMK*, *ITPK1*, *IP6K1*, *IPPK*, *IP6K2*, *PPIP5K1*, *PPIP5K2*, *IP6K3*, *IMPA1*, *IMPA2*, *INPP5B*, *OCRL*, *SYNJ1*, and *INPP5J*, while in tryptophan metabolism signaling, the genes with a high statistical significance were *ACAT1*, *ECHS1*, *GCDH*, *AANAT*, *AOC1*, *ALDH2*, *ALDH9A1*, *ALDH3A2*, *AOX1*, *ASMT*, *CAT*, *CYP1A1*, *CYP1A2*, *CYP1B1*, *DDC*, *HADH*, *IDO1*, *MAOB*, *TDO2*, *TPH1*, *KYNU*, *INMT*, *HAAO*, *AADAT*, *AFMID*, *ACMSD*, and *ACAT2*.

## 3. Discussion

In this study, we illustrated the putative restrictive effects of MCT2 expression on tumor growth and local invasiveness, and investigated gene expression differences TAMs when MCT2 expression was reduced compared to normal conditions. In TAMs, metabolic pathways that showed the more prominent differences included glucose and glutamate pathways, Acetyl-coA metabolic process, activation of immune response, B-cell activation and differentiation, cAMP-mediated signaling, and T-cell differentiation, and response to oxidative stress. In addition, the functional and taxonomic features of the gut microbiome during tumor development were evaluated via UPLC-MS and targeted amplicon sequencing. The relationships between the gut microbiome and fecal metabolome and plasma-related metabolites showed that the expression of MCT2 in the context of a lung TC1 tumor model imposes significant differences in these systems.

Monocarboxylate transporters are important regulators for cellular bioenergetics, and the transport of pyruvate and lactate across cellular membranes is an essential process in mammalian cells [28,29]. MCT2 is highly expressed in neurons, where it plays an important role in cellular energy metabolism and lactate shuttle [30,31]. Several studies have shown that lactate accumulation in human tumors, including cervical tumors, head and neck cancers, and rectal adenocarcinomas—tumors with metastatic spread—exhibited a wider range and significantly higher levels of lactate than non-metastatic tumors [32,33,34]. The lactate content can vary between individual tumors, even if the tumors are the same size, grade, or entity [33,34,35]. Here, we found that lactate tumor levels are significantly increased in MCT2 KO mice, suggesting that the utilization of lactate under a normal MCT2 expression may prevent enhanced glycolysis and the Warburg effect by improved lactate bioenergetics. Furthermore, high rates of lactate production or increased intra-tumoral lactate concentrations promote the acidification of TAMs, and lead to polarity changes and transformation into pro-tumoral macrophages [36]. As such, high glycolytic rates of cancer cells can lead to the increased generation of lactate and an accumulation of H+ ions, which ultimately induce an acidified tumor microenvironment [37]. The presence of MCT2 facilitates the relocation of lactic acid and enhances its utilization as an energy substrate, thereby reducing glycolysis and preventing pro-tumoral polarity changes in TAMs while preserving the innate immune cell function [5,38,39,40]. Furthermore, MCT2 has been reported to be located in the mitochondria with a role for mitochondrial metabolism, and inhibition of MCT2 suppresses colorectal cancer progression via the induction of mitochondrial dysfunction [41]. In the tumors harvested from the tamoxifen-induced MCT2 KO, we found that their mitochondria were disrupted compared to CO tumors, suggesting that these alterations in MCT2 expression may impose adverse effects on the tumor cell bioenergetics and promote the emergence of more aggressive cancer.

Macrophages are crucial drivers of tumor-promoting inflammation, and TAMs contribute to tumor progression at different levels by promoting genetic instability, nurturing cancer stem cells, supporting metastasis, and taming protective adaptive immunity [42]. In the tumor, TAMs are a key component of the local tumor mesenchymal−epithelial transition (TME) where they can contribute to tumor immune system evasion; suppress T-cell activity; and control cancer initiation, progression, and metastasis in a large number of different malignancies [43,44]. Cancer cells and TAMs co-exist in the context of a complex, bidirectional metabolic relationship that is not only dictated by, but also impinges upon the immunology of the TME [45,46]. Furthermore, TAMs can support tumor progression by (a) indirectly increasing the availability of selected nutrients in the TME, (b) providing trophic signals to malignant cells, and (c) mediating a robust immunosuppressive function [43]. In this study, we purified and performed bulk RNA-seq from TAMs derived from tumors in MCT2 KO and CO mice, and identified multiple metabolic pathways that are associated with tumor growth including, glucose and glutamate processes, T cells differentiation, and the Acetyl-coA metabolic process. For example, acetyl coenzyme A is a metabolite derived from several bioenergetics pathways (e.g., glycolysis, fatty acid oxidation, and amino-acid catabolism) and is further metabolized by the tricarboxylic acid cycle. Hypoxic TAMs within tumors can shift toward oxidative metabolism coupled with decreased glucose intake, culminating in endothelial cell activation, leading to neoangiogenesis and metastasis because of increased glucose availability in the TME [47]. Thus, glycolysis and higher lactate levels in TAMs and within the tumors themselves can support tumor growth, despite an increased competition for local glucose availability. In some murine models, these observations may reflect the requirement for glycolysis in M2 polarization [48].

Mitochondria have been implicated in the process of carcinogenesis, which includes alterations of the cellular metabolism and cell death pathways. Alterations of the mitochondrial networks are directly or indirectly involved in processes resulting in hypoxia-tolerant and hypoxia-sensitive gliomas, and by the hypoxia-inducible factor-1 (HIF-1), glycolytic protein isoforms, and fatty acid synthase [49]. In addition, the mitochondria in cancer cells have been observed with a lucent-swelling matrix associated with disarrangement and the distortion of cristae and partial or total cristolysis [49,50], supporting the presence of damaged mitochondria in cancers [51,52]. Mitochondrial changes are associated with mitochondrial-DNA mutations, tumoral microenvironment conditions, and mitochondrial fusion−fission disequilibrium [49]. In colorectal cancer cell lines, the knockdown of MCT2 causes mitochondrial dysfunction, cell-cycle arrest, and senescence without additional cellular stress [51].

It has become apparent that both the gut microbiome and plasma metabolome can play a role in the pathogenesis of tumor cells. Our study identified a number of associations between the altered gut microbiota and plasma metabolites, suggesting that alterations in the MCT2 expression in the host can promote the emergence of changes in the gut microbiota composition, which in turn may play a role in tumor growth and aggressiveness. Indeed, the gut microbiota has broad effects that contribute to host immune function at steady state and during tumorigenesis [16]. The gut microbiome and the immune system interact to maintain homeostasis of the gut, and alterations in the microbiome composition leads to immune dysregulation, promoting chronic inflammation and the development of tumors [53]. We believe that the microbiota differences associated with a reduced expression of MCT2 may reflect unique interactions between plasma metabolites and the gut microbiome which then foster further alterations in fecal and consequently plasma metabolomes. It is now clearly established that the gut microbiota is involved in the physiological activities of the host by affecting the bile acid pool, thus regulating hormone secretion and immunity via the resulting metabolites including the triggering of proinflammatory or immunosuppressive processes can be further affected by microorganisms [54,55]. Cancer patients seem to harbor a specific microbiome composition in the tumor niche, and disruption of the intestinal barrier function may trigger inflammation and carcinogenesis [56]. Microbial metabolites may interact directly with cancer cells or may regulate carcinogenesis by interacting with other components of TAMs, participating in immune responses or angiogenesis [57,58].

Since the gut microbiota interacts extensively with the host through substrate co-metabolism and metabolic exchange, we performed and analyzed concurrently obtained fecal and plasma metabolomes from the same animals. In our studies, at the phylum level, *Firmicutes and Bacteroidetes* accounted for more than 80% of the phylla, and the most abundant families included Lachnospiraceae, Marinifilaceae, and Ruminococcaceae, with most abundant genera being Lachnoclostridium and Ruminiclostridium. We found that Firmicutes increased in KO mice, but were drastically reduced in CO mice. In this context, *Firmicutes* and *Bacteroidetes* play an important role in regulating host energy metabolism. *Bacteroidetes* first oxidize pyruvate to acetyl-CoA, and then produce acetic acid via phospho-acetyltransferase and acetate kinase [59]. Furthermore, *Bacteroidetes* can also form propionic acid via the acrylic acid pathway, and there is a growing recognition of the importance of the *Firmicutes/Bacteroidetes* in metabolic diseases (Stearns et al., 2017). The significant increases in *Firmicutes/Bacteroidetes* values in KO mice support the assumption that tumor growth and invasion may be influenced by some of the changes in the gut microbiota.

As indicated in the results, metabolomic analyses identified significant differences in multiple metabolites and corresponding pathways that are relevant to cancer biology. As illustrative examples, the inositol phosphate metabolism pathway regulates cell proliferation, migration, and phosphatidylinositol-3-kinase (PI3K)/Akt signaling, and is frequently dysregulated in cancer [60], and tryptophan metabolism regulates the kynurenine pathway inducing element of the immune response [61]. Similarly, fatty acid synthase was identified as the tumor antigen OA-519 in aggressive breast cancer, and may play an importance role of fatty acid biosynthesis for cancer cell growth and survival [62,63]. In addition, the arachidonic acid pathway plays a key role in cardiovascular biology, carcinogenesis, and many inflammatory diseases [64], and promotes tumor progression [65,66]. Thus, the multifaceted effects of MCT2 span a complex network of metabolic pathways that regulate tumor proliferation and local invasiveness.

Among the functional aspects of MCT2 in host−tumor interactions, several questions on the role of MCT2 in TAMs remain unanswered. What are the molecular mechanisms underlying the ability of TAM to rapidly switch their metabolic and functional profile following the induced reduction of MCT2 expression? How does MCT2 affect the evolution of the TAM landscape during tumor progression? Would an overexpression of MCT2 or the administration of MCT2 analogs confer any benefit to the functional interplay between TAMs, immune responses, and tumor progression? We should point out that in the syngeneic model of TC1 cancer in mice, the tumor cells’ constitutive expression of MCT2 was preserved, such that the knockdown only affected the host, leaving the tumor cells unaffected. Future studies should examine the effects of knockdown or the overexpression of MCT2 in the tumor cells in mice with a normal expression of MCT2 or in KO mice.

In summary, our study strongly supports the concept that MCT2 confers beneficial effects in the host response to the tumor, and plays a role in tumor growth and tumor invasion, likely by regulating the concentrations of lactate and likely other monocarboxylates within the tumor. In addition, we showed that perturbations in the MCT2 expression induce substantial changes in gut microbiota, fecal, and plasma metabolites, all of which may play a role in tumor biology and immune responses.

## 4. Materials and Methods

### 4.1. Animals

The conditional MCT2 (Slc16a7) knockout C57BL/6 mouse model project #386-MCT-2 was generated at Ingenious Targeting Laboratory (Ronkonkoma, NY, USA). MCT2^loxP^ mice were intercrossed with mCre-Tg mice (The Jackson Laboratory stock # 008463) to generate MCT2^loxP/loxP^; Cre^+^ mice (MCT2 cKO). MCT2 cKO mice (8 week-old) were administered 0.18 mg of tamoxifen per gram of body weight via injection. Mice were dosed once daily for five consecutive days, and TC1 cells (100 K/mouse) were injected two days later. The MCT2^loxP^ mouse genotyping Primer: MCT2-R: CTA TCA CGC TGT TGC TGT AAG A; MCT2-F: GAC TCC CTT CTC CCA TCT CAG, wild type 319 bp, and Flox+/+ 380 bp. The schema for generating MCT2 mice knockout is shown in Appendix A. Cre+ genotyping of primer: 8463-1: AAA GTC GCT CTG AGT TGT TAT (mCre Wild type forward); 8463-2: GGA GCG GGA GAA ATG GAT ATG (mCre Wild type reverse); and 8463-3: CCT GAT CCT GGC AAT TTC G (mCre reverse). Cre+/+ = 825 bp, and Wild type = 650 bp. MCT2-positive males at 2 months of age were crossed with MCT-2 positive females at 2 months of age. The complete details about MCT2 knockout mice are presented in Appendix A. MCT2 injected with tamoxifen is called MCT2 KO and MCT2 injected with vehicle (WT) is called MCT2 CO. In the next section, MCT2 KO is called KO and MCT2 CO is CO.

### 4.2. MCT2 Mouse Genotyping

Animal breeding was carried out using MCT2 cKO male and female homozygous mice, and all offspring were genotyped at 21 days. DNA was isolated from the mouse tails, and PCR was performed with MCT2 specific primers. Appendix A confirms MCT2 conditional knockout at a molecular weight size at 380 bp.

### 4.3. Tumor Cell Line

Mouse epithelial lung tumor cells, TC1, (ATCC, CRL-2785) were purchased from American Type Culture Collection (Manassas, VA, USA) and were cultured at 37 °C, 95% air, 5% CO_2_ incubator. TC1 cells were maintained in an RPMI medium supplemented with 2 mM L-glutamine, 10 mM HEPES buffer, 1 mM sodium pyruvate, 0.1 mM non-essential amino acids, 100 U/penicillin/100 µg/mL streptomycin, 10% fetal bovine serum (FBS), and geneticin 0.4 mg/mL (Life Technologies, Grand Island, NY, USA).

### 4.4. Tamoxifen and Subcutaneous Flank Tumor Model

Tamoxifen was freshly prepared in sunflower oil at a concentration of 20 mg/mL, and 180 µg of tamoxifen/g body weight of mice were injected daily for five consecutive days. Mice treated with tamoxifen KO, *n* = 16, and those treated with sunflower oil without tamoxifen CO, *n* = 16, were inoculated with TC1 cells (1 × 10^5^ cells in 0.2 mL PBS; all cells <3 passages) by subcutaneous injection into the right lower flank. Tumor volumes were estimated every 3 days by externally measuring the length and width with an electronic caliper. After 24 days from tumor injection, the mice were sacrificed and blood was collected along with tumor surgical dissection and assessment of local invasion [67,68].

### 4.5. Western Blots

Several tissues, including testis, cortex, and visceral adipose tissue samples, were homogenized in a lysis buffer (50 mM Tris, pH 7.5, 0.4% NP-40, 150 mM NaCl, 10 mg/mL Aprotinin, 20 mg/mL Leupeptin, 10 mM EDTA, 1 mM Sodium orthovanadate, 100 mM Sodium Fluoride, Sigma-Milipore, St. Louis, MO, USA). Protein concentrations were measured using the BCA kit (Life Technologies, Grand Island, NY, USA). Equal amounts of total protein from each tissue were electrophoresed using SDS-PAGE gel (4–20%) and were transferred into a nitrocellulose membrane (Millipore, Billerica, MA, USA). Following membrane transfer, incubation in a blocking buffer (5% nonfat dry milk in 25 mM Tris, pH 7.4, 3.0 mM KCl, 140 mM NaCl, and 0.05% Tween 20 (TBST)) for 1 h at room temperature was performed. Membranes were then incubated overnight at 4 °C with MCT2 polyclonal antibody from Bioss Antibodies Inc (# bs-3995R, Woburn, MA, USA. Membranes were washed with TBS-T, and incubated with horseradish peroxidase linked, conjugated β-Actin (Cat# 7074, RRID:AB_2099233, Cell Signaling Technology, Danvers, MA, USA) for 1 h at room temperature. Immunoreactive bands were visualized using an enhanced chemiluminescence detection system (Chemidoc XRS+; Bio-Rad, Hercules, CA, USA).

### 4.6. Transmission Electron Microscopy and Mitochondrial Dysfunction

Tumor tissues were dissected and processed by transmission electron microscopy (TEM). Tissues were fixed in 2% paraformaldehyde, 2% glutaraldehyde in 100 mM sodium cacodylate buffer pH = 7.35. Next, the fixed tissues were rinsed with 100 mM sodium cacodylate buffer, pH 7.35 containing 130 mM sucrose. Secondary fixation was performed using 1% osmium tetroxide (Ted Pella, Inc. Redding, CA, USA) in a cacodylate buffer using a Pelco Biowave (Ted Pella, Inc. Redding, CA, USA) operated at 100 Watts for 1 min. Specimens were next incubated at 4 °C for 1 h, then rinsed with a cacodylate buffer and further with distilled water. En bloc staining was performed using 1% aqueous uranyl acetate and incubated at 4 °C overnight, then rinsed with distilled water. A graded dehydration series was performed using ethanol at 4 °C, transitioned into acetone, and dehydrated tissues were then infiltrated with Epon resin for 24 h at room temperature and polymerized at 60 °C overnight. Sections were cut to a thickness of 85 nm using an ultramicrotome (Ultracut UCT, Leica Microsystems, Wetzlar, Germany) and a diamond knife (Diatome, Hatfield, PA, USA). Images were acquired with a JEOL JEM 1400 transmission electron microscope (JEOL, Peabody, MA, USA) at 80 kV on a Gatan Ultrascan 1000 CCD (Gatan, Inc, Pleasanton, CA, USA). Unless otherwise indicated, all reagents were purchased from Electron Microscopy Sciences and all specimen preparations were performed at the Electron Microscopy Core Facility, University of Missouri.

### 4.7. Lactate Measurement

The tumors were weighted and homogenized in PBS and adjusted to yield the same concentration of tissue in each sample. Samples were then centrifuged at 13,000× *g* for 10 min, the supernatants were collected, and the protein contents were measured. Lactate quantifications were performed at room temperature with the Lactate Assay Kit (BioVision, K607-100, Milpitas, CA, USA) following the manufacturer’s protocol. Lactate concentrations were normalized with the protein concentration levels in each sample.

### 4.8. Isolation Tumor Macrophage

Subcutaneous tumors were dissected from KO and CO mice and tumors were transferred into cold 0.1% BSA in an RPMI medium (Life Technologies, Grand Island, NY, USA). Tumors were minced into small pieces and digested with 2 mg/mL collagenase type 4 in 0.1% BSA in RPMI medium (1 g tumor/10 mL medium). The tumors were further at incubated at 37 °C with gentle shaking for 45 min. The tumor cells were filtered through a 70 μm strainer and centrifuged at 2000× *g* for 5 min. The cells were washed twice with full medium (RPMI1640 with 10% FBS) and were suspended at a concentration of 1 × 10^8^/mL in RPMI1640. D11b+ cell from cell suspension was isolated according to the manufacturer protocol of EasySep Mouse CD11b Positive Selection Kit II (STEMCELL, # 18970, Vancouver, BC, Canada).

### 4.9. RNA-seq Analysis of Tumor Macrophages

Tumor-associated macrophages (TAMs) were purified from digested tumors by CD11b-PE magnetic labelling (EasySepTMMouse CD11b Positive Selection Kit, StemCell Technologies, Vancouver, BC, Canada) following the manufacturer’s protocol. The total RNAs were isolated from the TAMs from both KO and CO mice using the RNeasy Tissue Mini Kit (Qiagen, Valencia, CA, USA), as described [69]. The total RNA quality and integrity were assessed using the Eukaryote Total RNA Nano 6000 LabChip assay (Agilent Technologies, Santa Clara, CA, USA) on an Agilent 2100 Bioanalyzer. The total RNA samples were quantified by measuring A260 nm on a UV/VIS spectrophotometer (ND-1000, NanoDrop Technologies, Wilmington, DE, USA). Poly-A enriched mRNASeq libraries were prepared following Illumina’s TruSeq Stranded mRNA LT library preparation protocol (Illumina Inc., San Diego, CA, USA) using 1 μg of total RNA. The cDNA sequencing libraries were generated from poly-A selected RNA using a TrueSeq library preparation kit (Illumina, Inc., San Diego, CA, USA). All of the sequencing was performed on an Illumina HiSeq 2500 (Illumina, Inc., San Diego, CA, USA).

The raw RNA-seq data were analyzed using the FastQ Screen Trimmomatic to remove the adaptor and low-quality sequences, and the filtered reads were mapped to GRCm38 with HISAT2 [70]. The expression level of each gene was quantified as FPKM https://toppgene.cchmc.org/ (fragments per kilobase of exon per million mapped fragments, accessed on 5 May 2021) and counts, and the DESeq2 algorithm (http://cole-trapnell-lab.github.io/cufflinks/install/, accessed on 8 June 2021) was applied to filter the different expression genes (DEGs). The significance of the differentially expressed genes was identified based on the adjusted raw *p*-values to a false discovery rate (FDR) of <0.05 and fold changes (log2 FC > 1). Gene ontology (GO, http://www.geneontology.org/ (accessed on 16 April 2021)) and Kyoto encyclopedia of genes and genomes (KEGG, http://www.genome.jp/kegg/analyses (accessed on 13 March 2021)) were explored to evaluate the biological function of the DEGs [71].

### 4.10. Functional Enrichment Analysis of DEGs

Gene set enrichment analysis (GSEA) was used to rank all of the genes in the dataset based on differential expression. A total of 1000 permutations were performed to estimate the empirical *p*-values for the gene sets. Details of GSEA can be found in Subramanian et al. [72]. For understanding the biological processes and pathways in which the DEGs are involved, GO (Gene Ontology) and KEGG (Kyoto Encyclopedia of Genes and Genomes) enrichment analyses were performed on the DAVID database. The construction and analysis of the protein−protein interaction (PPI) network was performed using the STRING database (http://www.string-db.org/, accessed on 22 June 2021) for a network of DEGs with high confidence (confidence score > 0.70).

### 4.11. DNA Extraction

Feces from MCT2 KOKO and CO mice underwent DNA extraction using PowerFecal kits (Qiagen) according to the manufacturer’s instructions, with the exception that samples were homogenized in the provided bead tubes using a TissueLyser II (Qiagen, Venlo, The Netherlands) for three minutes at 30 s, rather than performing the initial homogenization of samples using the vortex adapter described in the protocol. Samples were then eluted in 100 µL of elution buffer (Qiagen). DNA yields were quantified via fluorometry (Qubit 2.0, Invitrogen, Carlsbad, CA, USA) using quant-iT BR dsDNA reagent kits (Invitrogen) and normalized to a uniform concentration and volume.

### 4.12. 16S rRNA Library Preparation and Sequencing

Extracted fecal DNA was processed at the University of Missouri DNA Core Facility. Bacterial 16S rRNA amplicons were constructed via amplification of the V4 region of the 16S rRNA gene with universal primers (U515F/806R) previously developed against the V4 region, flanked by Illumina standard adapter sequences [73,74]. Oligonucleotide sequences are available at proBase [75]. Dual-indexed forward and reverse primers were used in all reactions. PCR was performed in 50 µL reactions containing 100 ng metagenomic DNA, primers (0.2 µM each), dNTPs (200 µM each), and Phusion high-fidelity DNA polymerase (1U, Thermo Fisher). Amplification parameters were 98 °C^(3 min)^ + [98 °C^(15 s)^ + 50 °C^(3 s)^ + 72 °C^(3 s)^] × 25 cycles + 72 °C^(7 min)^. Amplicon pools (5 µL/reaction) were combined, thoroughly mixed, and then purified by the addition of Axygen Axyprep MagPCR clean-up beads to an equal volume of 50 µL of amplicons and incubated for 15 min at room temperature. The products were then washed multiple times with 80% ethanol and the dried pellet was resuspended in 32.5 µL EB buffer (Qiagen), incubated for 2 min at room temperature, and then placed on the magnetic stand for 5 min. The final amplicon pool was evaluated using the Advanced Analytical Fragment Analyzer automated electrophoresis system, quantified using quant-iT HS dsDNA reagent kits, and diluted according to Illumina’s standard protocol for sequencing on the MiSeq instrument.

### 4.13. Informatics Analysis

DNA sequences were assembled and annotated at the MU Informatics Research Core Facility. Primers were designed to match the 5′ ends of the forward and reverse reads. Cutadapt [76] (version 2.6; https://github.com/marcelm/cutadapt, accessed on 6 January 2021) was used to remove the primer from the 5′ end of the forward read. If found, the reverse complement of the primer to the reverse read was then removed from the forward read, as were all bases downstream. Thus, a forward read could be trimmed at both ends if the insert was shorter than the amplicon length. The same approach was used on the reverse read, but with the primers in the opposite roles. Read pairs were rejected if one read or the other did not match a 5′ primer, and an error-rate of 0.1 was allowed. Two passes were made over each read to ensure removal of the second primer. A minimal overlap of three bp with the 3′ end of the primer sequence was required for removal.

The QIIME2 [77] DADA2 [78] plugin (version 1.10.0) was used to denoise, de-replicate, and count ASVs (amplicon sequence variants), incorporating the following parameters: (1) forward and reverse reads were truncated to 150 bases, (2) forward and reverse reads with number of expected errors higher than 2.0 were discarded, and (3) Chimeras were detected using the “consensus” method and removed. R version 3.5.1 and Biom version 2.1.7 were used in QIIME2. Taxonomies were assigned to final sequences using the Silva.v132 [79] database, using the Classify-sklearn procedure.

The raw data were deposited in NIH with submission ID: SUB9574437, and BioProject ID: PRJNA726916. https://www.ncbi.nlm.nih.gov/bioproject/726916, accessed on 5 January 2021.

### 4.14. Metabolomic Profiling

For fecal metabolomics, 10 mg of each fecal sample was used and 1.0 mL of 80% methanol containing 18 µg/mL of umbelliferone (Sigma-Milipore, St. Louis, MO, USA) was added followed by sonication, and was then vortexed for 20 s each. The samples were then shaken in an orbital shaker for 2 h at 140 rpm and centrifuged at 3000× *g* for 40 min.

For plasma metabolomics, 100 µL of each plasm sample were used, and 1.0 mL of 80% methanol containing 18 µg/mL of umbelliferone and 0.1% formic acid were added, followed by sonication and were then vortex for 20 s each. Samples were then shaken in an orbital shaker for 2 h at 140 rpm and centrifuged at 3000× *g* for 40 min.

Supernatant of 0.5 mL was transferred to an autosampler vial and dried under nitrogen followed by reconstitution in 100 µL for the LCMS analysis. LCMS data were acquired using a waters aquity UHPLC system coupled with a Bruker Impact II QTOF mass spectrometer. Data extraction and normalization were performed using Bruker’s Metaboscape 4.0 software and statistical analysis were conducted using MetaboAnalyst4 software.

To each of the remaining sample solutions, 1.5 mL of CHCl3 containing 10 µL/mL of docosanol was added, then sonicated and vortexed for 20 s each followed by incubation at 50 °C for 1 h. Then, 1 mL of HPLC grade water containing 25 µg/mL of ribitol was added to each sample, sonicated, and vortexed for 20 s each and incubated for 1 h at 50 °C. Sample tubes were centrifuged at 3000× *g* for 40 min and allowed to stand for 5 min. Then, 1 mL solution from the upper layer non-polar GCMS analysis and another 1 mL solution from the bottom layer for polar GCMS analysis were transferred to two separate autosampler vials. One pooled sample was prepared each for polar and non-polar GCMS by combining 10 µL of solution drawn from each sample. All of the solutions in the autosampler vials were dried using a gaseous nitrogen stream.

For polar GC−MS analysis, dried samples were methoximated in pyridine with 50 μL of 15 mg/mL methoxyamine hydrochloride, and then trimethylsilylated with 50 μL MSTFA (N-methyl-N-(trimethyl-silyl)trifluoroacetamide) + 1%TMCS (chlorotrimethylsilane) reagent. Samples for non-polar GCMS analysis were reconstituted in 50 µL of pyridine followed by trimethylsilylation with 50 µL of MSTFA + 1% TMCS. The derivatized extracts were then analyzed for non-targeted metabolic profiling using an Agilent 6890 GC coupled to a 5973 N MSD mass spectrometer with a scan range from m/z 50 to 650 (Agilent Technologies, Inc., Santa Clara, CA, USA). Then, 1 µL of sample was injected into the GC column with a split ratio of 1:5 for polar GCMS and 1:1 for non-polar GCMS analysis. Separation was achieved with a temperature program of 80 °C for 2 min, then ramped at 5 °C/min to 315 °C and held at 315 °C for 12 min, a 60 m DB-5MS column (J&W Scientific, 0.25 mm ID, 0.25 μm film thickness) and a constant flow of 1.0 mL/min of helium gas. A standard alkane mix was used for GCMS quality control and retention index calculations. The data from the pooled sample were deconvoluted using AMDIS and annotated through mass spectral and retention index matching to an in-house constructed spectra library. The unidentified components were then searched and identified using spectral matching to a commercial NIST17 mass spectral library. The combined identifications were saved as an. ELU file, and the abundance of the ions in all the other samples were extracted using custom MET-IDEA software. The abundances were then normalized to the internal standard, ribitol, and the normalized values were used for statistical comparisons using Metaboanalyst4 software.

The supernatant (0.5 mL) was transferred to an autosampler vial for liquid chromatography−mass spectrometry (LC−MS) analysis, wherein data were acquired using a Waters Aquity UHPLC system coupled with a Bruker Impact II QTOF mass spectrometer. Data extraction and normalization were performed using Bruker’s Metaboscape 4.0 software and statistical analysis (https://www.bruker.com/products/mass-spectrometry-and-separations/ms-software/metaboscape.html, accessed on 12 February 2021) and MetaboAnalyst4 software (https://www.metaboanalyst.ca/, accessed on 6 February 2021). To each of the remaining sample solutions, 1.5 mL of CHCl3 containing 10 µL/mL of docosanol was added, then sonicated and vortexed for 20 s each followed by incubation at 50 °C for 1 h. One mL of HPLC grade water containing 25 µg/mL of ribitol was added to each sample, sonicated and vortexed for 20 s each, and incubated for 1 h at 50 °C. The sample tubes were centrifuged at 3000× *g* for 40 min and allowed to stand for 5 min. The upper layer non-polar (1 mL) for gas chromatography−mass spectrometry (GC−MS) non-polar analysis, and another 1ml solution from the bottom layer for the polar GC−MS analysis were transferred to two separate autosampler vials. One pooled sample was prepared each for polar and non-polar GC−MS by combining 10 µL of solution drawn from each sample. All the solutions were dried in autosampler vials using gaseous nitrogen stream. Dried samples for polar GC−MS analysis were methoximated in pyridine with 50 μL of 15 mg/mL methoxyamine hydrochloride, and then trimethylsilylated with 50 μL MSTFA (N-methyl-N-(trimethyl-silyl) trifluoroacetamide) + 1% TMCS (chlorotrimethylsilane) reagent. The samples for non-polar GC−MS analysis were reconstituted in 50 µL of pyridine followed by trimethylsilylation with 50 µL of MSTFA + 1% TMCS. The derivatized extracts were then analyzed for non-targeted metabolic profiling using an Agilent 6890 GC coupled to a 5973 N MSD mass spectrometer with a scan range from m/z 50 to 650 (Agilent Technologies, Inc., Santa Clara, CA, USA). Then, 1 µL of sample was injected into the GC column with a split ratio of 1:5 for polar GC−MS and 1:1 for non-polar GC−MS analysis. Separation was achieved with a temperature program of 80 °C for 2 min, then ramped at 5 °C/min to 315 °C and held at 315 °C for 12 min, a 60 m DB-5MS column (J&W Scientific, 0.25 mm ID, 0.25 μm film thickness) and a constant flow of 1.0 mL/min of helium gas. A standard alkane mix was used for the GC−MS quality control and retention index calculations. The data from the pooled sample were deconvoluted using AMDIS and were annotated through mass spectral and retention index matching to an in-house constructed spectra library. The unidentified components were then searched and identified using spectral matching to a commercial NIST17 mass spectral library. For the non-targeted ultra-high performance liquid chromatography tandem mass spectrometry (UHPLC-MS/MS), characterization of fecal samples was performed as described in [80,81,82]. For the UHPLC methods, samples were shaken in an orbital shaker for 2 h at 140 rpm and centrifuged at 3000× *g* for 40 min, and 0.5 mL of the supernatant was transferred to an autosampler vial for LCMS analysis wherein the data were acquired using a waters acquity UHPLC system coupled with a Bruker Impact II QTOF mass spectrometer. The combined identifications were saved as an ELU file, and the abundance of the ions in all the other samples were extracted using custom MET-IDEA software. The abundances were then normalized to the internal standard, ribitol, and the normalized values were used for statistical comparisons using Metaboanalyst4 software.

### 4.15. Statistical Analysis

Differences in OTU relative abundance between KO and CO samples were determined using Student’s t-test. The significance of the differences between the means of the groups was compared by ANOVA using the Statistical Package (version 21.0, SPSS Inc., Chicago, IL, USA). Two-way ANOVA with the Student Newman−Keuls post-hoc method was used to assess differences in MCT2 groups, where *p*  <  0.05 was considered statistically significant. Data show the mean of independent biological experiments with the standard deviation (± SD), unless otherwise indicated. Multivariate statistical analyses such as ANOVA, box plots, and Volcano plots were performed with the MetaboAnalyst 3.0 program after data pre-treatments, i.e., normalization to the sum, log transformation, and Pareto scaling. Changes in metabolite abundances were considered statistically significant at *p* < 0.05. To account for quantitative and qualitative community differences between groups, statistical testing for β-diversity was performed via a two-way PERMANOVA analysis of both Bray−Curtis and Jaccard dissimilarities for bacterial OTU community structure.

## Figures and Tables

**Figure 1 ijms-22-10616-f001:**
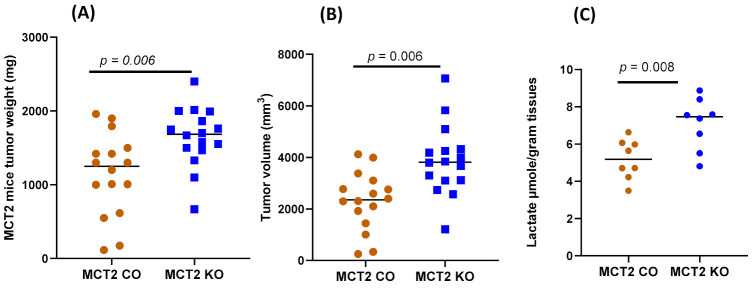
Individual TC1 tumor weight, volume, and lactate levels in MCT2 KO and CO mice: (**A**) tumor weight; (**B**) tumor volume; (**C**) tumor lactate concentrations.

**Figure 2 ijms-22-10616-f002:**
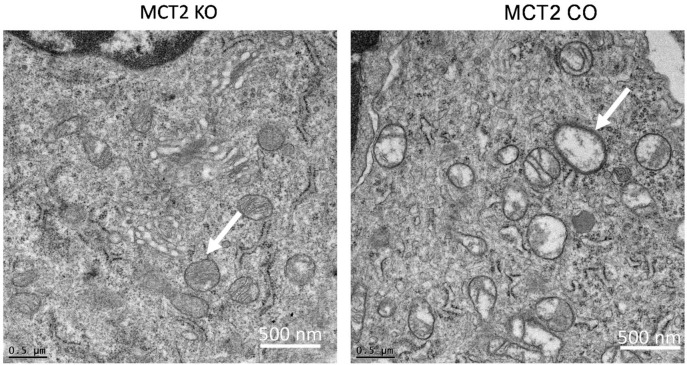
Representative electron microscopy images of mitochondria in the tumor tissues of KO and CO mice. Arrows indicate examples of disrupted mitochondria. *n* = 6/experimental group.

**Figure 3 ijms-22-10616-f003:**
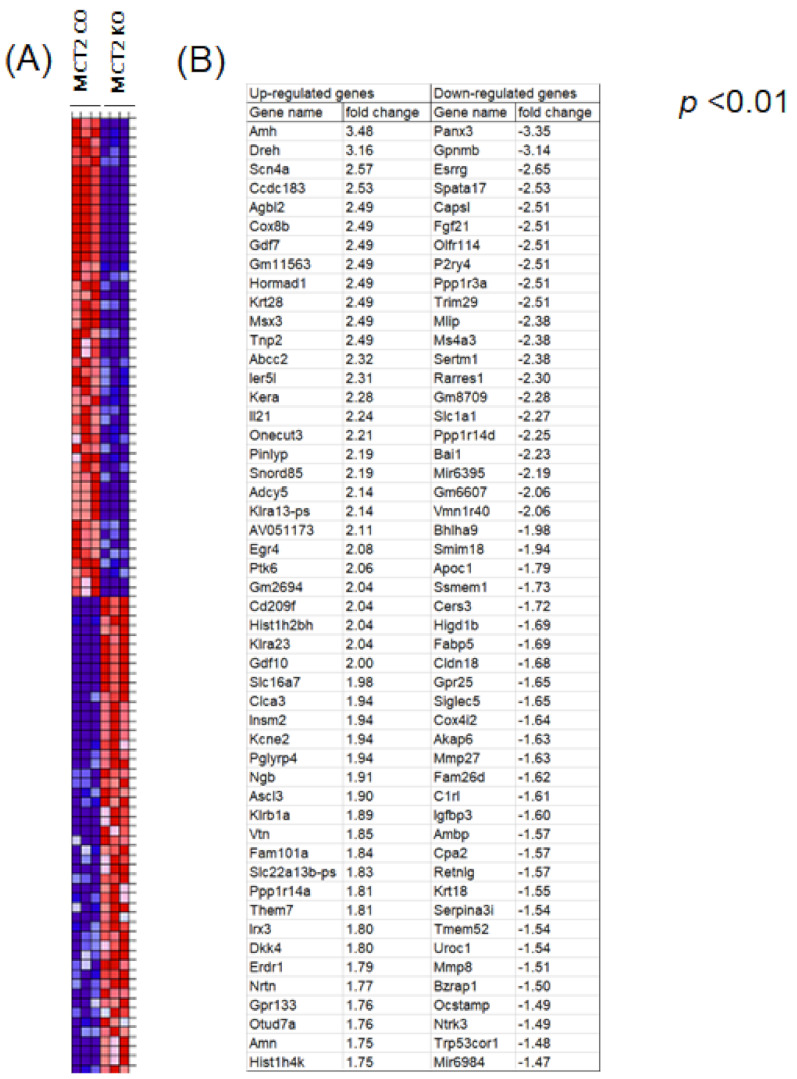
Cluster analysis of differential gene expression in tumor associated macrophages within TC1 tumors in MCT2 KO and CO mice. Heatmap of the top 50 up-regulated (**A**) and top 50 down-regulated genes (**B**). Expression values for each gene (row) are normalized across all samples (columns) by Z-score. Both column and row clustering were applied, and distinct gene clusters identified by the Gap statistic method are shown to illustrate the major expression patterns observed in the data.

**Figure 4 ijms-22-10616-f004:**
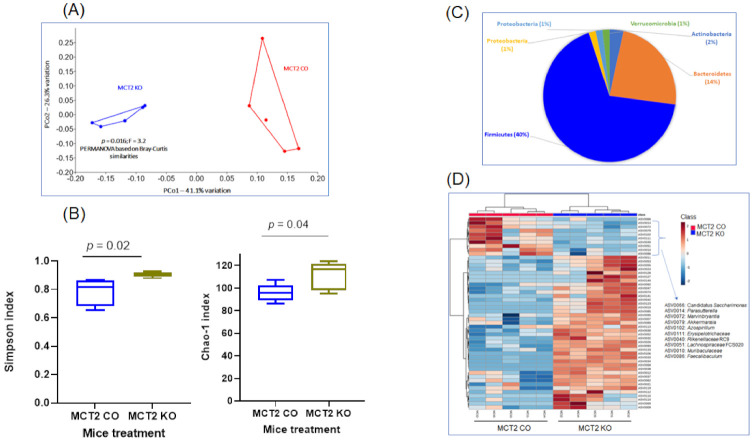
Fecal bacterial composition of 16S rRNA and operational taxonomic units (OTU) identified in MCT2 KO and CO mice. (**A**) Principal coordinate analysis (PCoA) plot ordinated by Bray−Curtis beta-diversity similarities for fecal microbiota. Red indicates CO and blue indicates KO samples. PC—principal component; PERMANOVA—permutational multivariate analysis of variance, *p* = 0.016, F = 3.2. (**B**) Pie chart showing the relative abundance of Phylum detected in KO and CO mice. (**C**) Comparisons of gut microbial alpha-diversity and beta-diversity between CO and KO as estimated the Chao1 index and by the Shannon index. (**D**) Hierarchical cluster analysis of the 50 consistently detected OTUs in the gut microbiota (*n* = 5/group).

**Figure 5 ijms-22-10616-f005:**
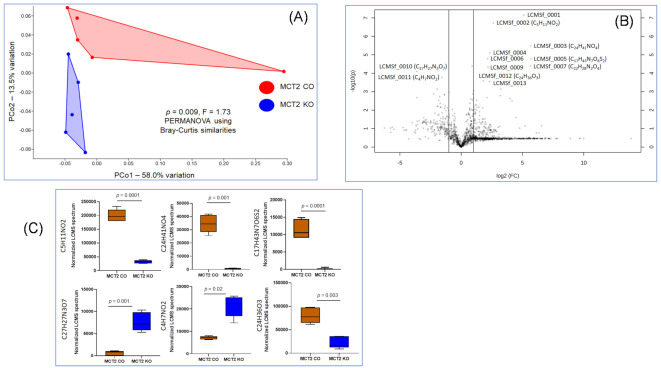
Fecal metabolome for MCT2 KO and CO mice. Fecal metabolome was identified via ultra-high-pressure liquid chromatography−mass spectroscopy (UHPLC−MS). (**A**) PCoA plot ordinated by Bray−Curtis beta-diversity similarities for fecal metabolome, red indicates KO and blue indicates CO samples. PC—principal component; PERMANOVA—permutational multivariate analysis of variance *p* = 0.0009, F = 1.73. (**B**) Volcano plot showing fold difference (FD, *x*-axis) and *p*-value (*y*-axis) associated with metabolites detected at significantly greater relative abundance in MCT2 KO and CO. (**C**) Box plots for six highly significant metabolites identified in the Volcano plot (*n* = 5/condition).

**Figure 6 ijms-22-10616-f006:**
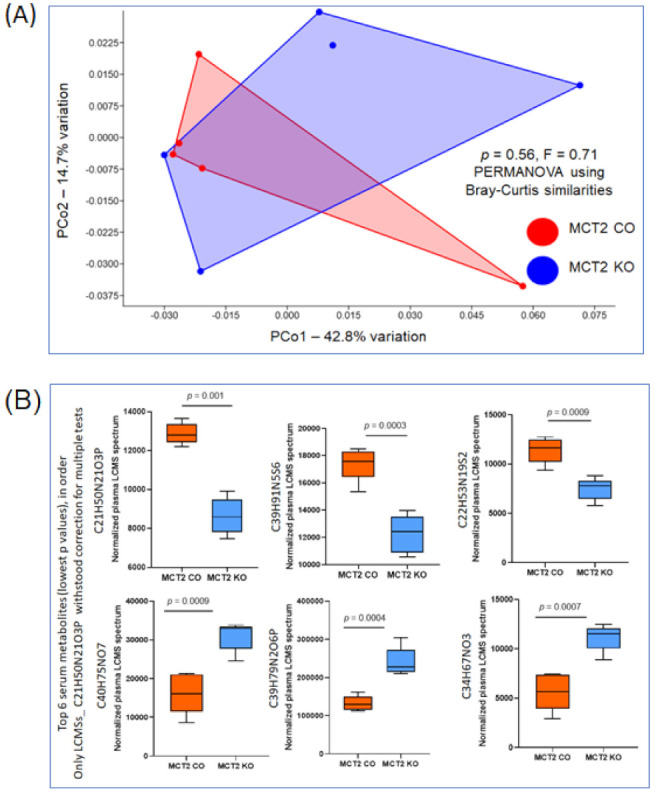
Plasma metabolome for MCT2 KO and CO. Plasma metabolomes were identified via ultra-high-pressure liquid chromatography−mass spectroscopy (UHPLC−MS). (**A**) PCoA plot ordinated by Bray−Curtis beta-diversity similarities for plasma metabolome, red color indicates CO and blue color indicates KO samples. PC—principal component; PERMANOVA—permutational multivariate analysis of variance *p* = 0.56, F = 0.71, *n* = 5. No significant differences were identified in the Volcano plot. (**B**) Box plots for the top six plasma metabolites (lowest *p* values) after correction for multiple tests (*n* = 5/condition).

**Figure 7 ijms-22-10616-f007:**
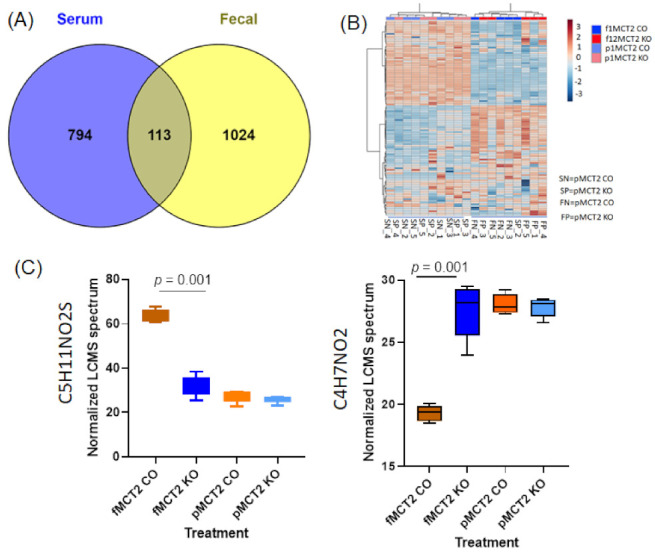
Comparisons of fecal and plasma metabolites identified using UHPLC-MS in MCT2 KO and CO. (**A**) Venn diagram for fecal and plasma metabolites showing the unique metabolites for each and the interactions among them. (**B**) Hierarchical clustering of samples based on the relative abundance of the interaction between fecal and plasma samples for MCT2 KO and CO. (**C**) Box plots showing the two interaction metabolites between the plasm and fecal samples among 113 common LCMS compounds, based on two-way ANOVA: box plots showing the distribution in each group of the two metabolites with significant treatment-associated differences.

**Figure 8 ijms-22-10616-f008:**
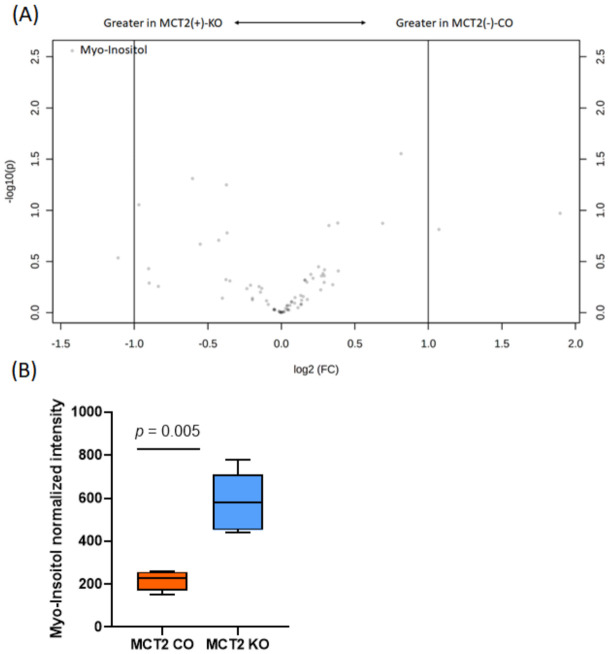
Plasma polar gas chromatography−mass spectroscopy (GC−MS). (**A**) Volcano plot showing the fold difference (FD, *x*-axis) and *p*-value (*y*-axis) associated with polar metabolites. (**B**) Tukey box plots showing the differences in abundance of Myo-Inositol in MCT2 KO and CO mice.

**Figure 9 ijms-22-10616-f009:**
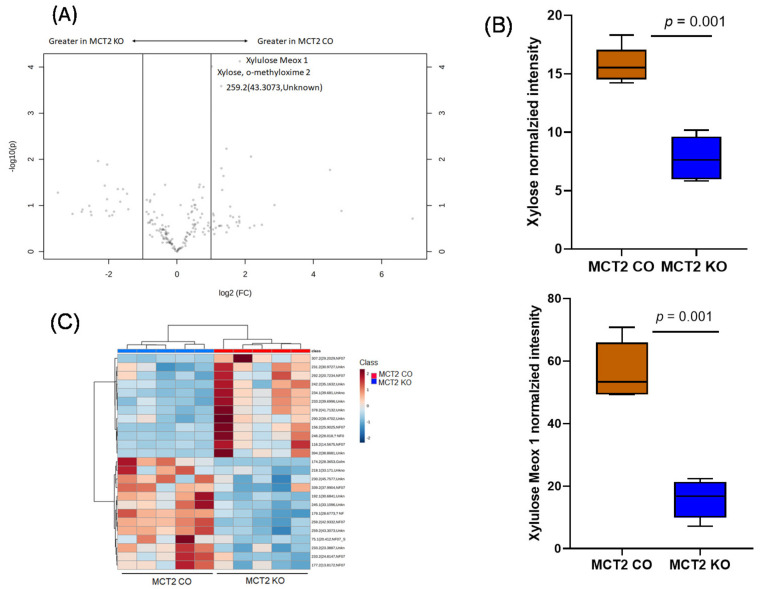
Fecal polar gas chromatography−mass spectroscopy (GC−MS). (**A**) Volcano plot showing fold difference (FD, *x*-axis) and *p*-value (*y*-axis) associated with polar metabolites. (**B**) Tukey box plots showing the significant abundance of Xyloses, Xylose, and unknown metabolites in MCT2 KO and CO mice. (**C**) Hierarchical clustering of samples based on the relative abundance of the ASVs yielding the lowest *p*-values following ANOVA of all ASVs comparing MCT2 KO and CO mice.

**Figure 10 ijms-22-10616-f010:**
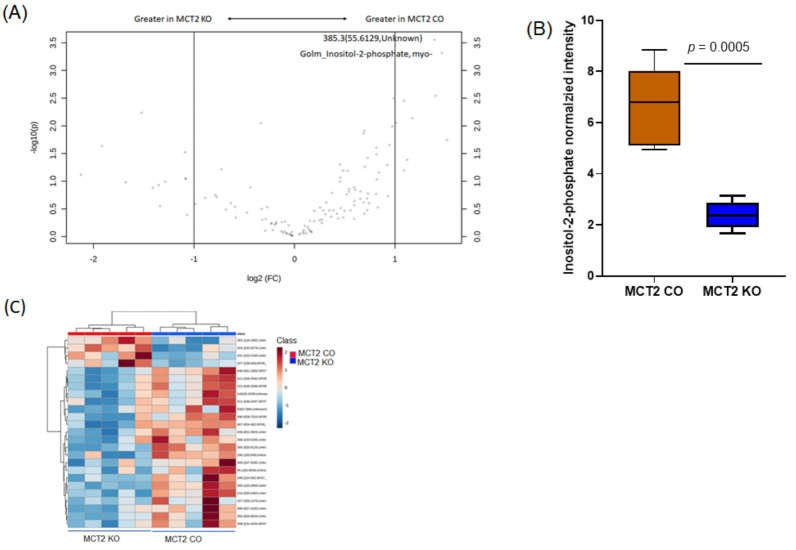
Fecal non-polar gas chromatography−mass spectroscopy (GC−MS). (**A**) Volcano plot showing fold difference (*x*-axis) and *p*-value (*y*-axis) associated with polar metabolites. (**B**) Tukey box plots showing the significant abundance of five metabolites in MCT2 KO and CO mice. (**C**) Hierarchical clustering of samples based on the relative abundance of the ASVs.

**Figure 11 ijms-22-10616-f011:**
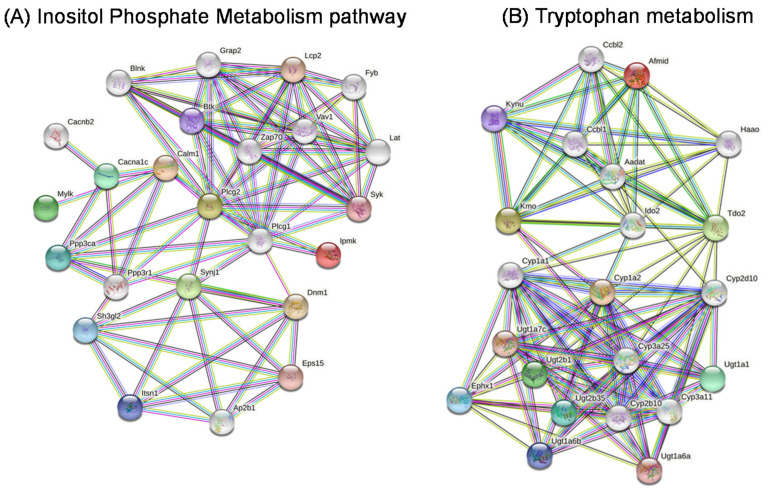
Gene network representation of the Inositol Phosphate Metabolism pathway (**A**) and Tryptophan metabolism (**B**) in MCT2 KO and CO mice.

**Table 1 ijms-22-10616-t001:** Gene ontology (GO) enrichment analysis of identified in top 50 upregulated and top 50 down regulated differentially expressed genes in tumor macrophages derived from KO compared to controls using RNA-seq.

GO Enrichment Analysis	#Term ID	Term Description	False Discovery Rate
**Molecular function**	GO:0031994	insulin-like growth factor I binding	1.5 × 10^−10^
	GO:0031995	insulin-like growth factor II binding	2.9 *×* 10^−9^
	GO:0005158	insulin receptor binding	5.1 *×* 10^−4^
	GO:0005159	insulin-like growth factor receptor binding	3.1 *×* 10^−3^
	GO:0008083	growth factor activity	2.5 *×* 10^−2^
	GO:0004859	phospholipase inhibitor activity	2.9 *×* 10^−2^
	GO:0005515	protein binding	2.9 *×* 10^−2^
	GO:0003677	DNA binding	4.2 *×* 10^−2^
	GO:0098772	molecular function regulator	4.2 *×* 10^−2^
	GO:0005160	transforming growth factor beta receptor binding	4.9 *×* 10^−2^
	GO:0005178	integrin binding	4.9 *×* 10^−2^
**Cellular Component**	GO:0005615	extracellular space	6.5 *×* 10^−7^
	GO:0000786	nucleosome	3.8 *×* 10^−6^
	GO:0000788	nuclear nucleosome	7.6 *×* 10^−5^
	GO:0005576	extracellular region	1.5 *×* 10^−4^
	GO:0000785	chromatin	2.1 *×* 10^−5^
	GO:0000790	nuclear chromatin	2.3 *×* 10^−5^
	GO:0000228	nuclear chromosome	1.6 *×* 10^−3^
	GO:0042568	insulin-like growth factor binary complex	2.1 *×* 10^−3^
	GO:0035867	alphav-beta3 integrin-IGF-1-IGF1R complex	4.9 *×* 10^−3^
	GO:0005694	chromosome	1.2 *×* 10^−2^
	GO:0001518	voltage-gated sodium channel complex	2.5 *×* 10^−2^
	GO:0005751	mitochondrial respiratory chain complex IV	4.4 *×* 10^−2^
**Biological process**	GO:0043567	regulation of insulin-like growth factor receptor	1.4 *×* 10^−7^
	GO:0043568	positive regulation of insulin-like growth factor receptor	4.6 *×* 10^−4^
	GO:0034728	nucleosome organization	5.1 *×* 10^−4^
	GO:0001649	osteoblast differentiation	1.4 *×* 10^−3^
	GO:0006323	DNA packaging	1.4 *×* 10^−3^
	GO:0014910	regulation of smooth muscle cell migration	1.4 *×* 10^−3^
	GO:0019556	histidine catabolic process to glutamate and formamide	1.4 *×* 10^−3^
	GO:0019557	histidine catabolic process to glutamate and formate	1.4 *×* 10^−3^
	GO:0042246	tissue regeneration	1.4 *×* 10^−3^
	GO:0006325	chromatin organization	1.6 *×* 10^−3^
	GO:0006333	chromatin assembly or disassembly	1.8 *×* 10^−3^
	GO:0010906	regulation of glucose metabolic process	1.8 *×* 10^−3^
	GO:0090031	positive regulation of steroid hormone biosynthetic process	1.8 *×* 10^−3^
	GO:0048009	insulin-like growth factor receptor signaling pathway	2.7 *×* 10^−3^
	GO:0045725	positive regulation of glycogen biosynthetic process	4.3 *×* 10^−3^

**Table 2 ijms-22-10616-t002:** KEGG pathways identified in top 50 upregulated and top 50 down regulated differentially expressed genes in tumor macrophages derived from KO compared with controls using RNA-seq.

KEGG ID	Term Description	False Discovery Rate
mmu00340	Histidine metabolism	0.0066
mmu04115	p53 signaling pathway	0.0066
mmu04610	Complement and coagulation cascades	0.0066
mmu05202	Transcriptional misregulation in cancer	0.0066
mmu05215	Prostate cancer	0.0066
mmu04350	TGF-beta signaling pathway	0.0091
mmu01522	Endocrine resistance	0.0104
mmu05322	Systemic lupus erythematosus	0.0104
mmu04066	HIF-1 signaling pathway	0.0122
mmu05205	Proteoglycans in cancer	0.0187
mmu04068	FoxO signaling pathway	0.0244

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
