# Peer review of "Monocarboxylate Transporter-2 Expression Restricts Tumor Growth in a Murine Model of Lung Cancer: A Multi-Omic Analysis"

_ijms, 2021, doi:10.3390/ijms221910616_

Round 1
Reviewer 1 Report
Dear Authors,
your manuscript sounds interesting and good. I would also a little improve Introduction and more detailed descriptions of the results.
However a minor revision is necessary.
Author Response
Reviewer 1
Comments and Suggestions for Authors
Dear Authors,
your manuscript sounds interesting and good. I would also a little improve Introduction and more detailed descriptions of the results.
However a minor revision is necessary.
We thank the reviver for the comments and suggestions, and we edited the introduction as the reviewer suggested.
A multi-omics approaches revealed a distinct tumor immune microenvironment contributing to immunotherapy in lung adenocarcinoma (Huang et al., 2021).
Huang Z, Li B, Guo Y, Wu L, Kou F, Yang L. Signatures of Multi-Omics Reveal Distinct Tumor Immune Microenvironment Contributing to Immunotherapy in Lung Adenocarcinoma. Front Immunol. 2021 Sep 3;12:723172. doi: 10.3389/fimmu.2021.723172. eCollection 2021.PMID: 34539658.
We also edited some of Results section namely MCT2 immunoreactivity.
Compared to vehicle-treated mice, tamoxifen-treated mice showed significant reductions in MCT2 protein expression in testis P<0.001, visceral fat P<0.01, and cortex P<0.03, respectively, Figure S1).
Reviewer 2 Report
Authors have employed multiple approaches to connect MCT2 and lung cancer. More explanation in the discussion is required to connect the effect of MCT2 on the drastic changes in the morphology of Mitochondria, especially the cristeolysis.
Author Response
Reviewer 2
Comments and Suggestions for Authors
Authors have employed multiple approaches to connect MCT2 and lung cancer. More explanation in the discussion is required to connect the effect of MCT2 on the drastic changes in the morphology of Mitochondria, especially the cristolysis.
We thank the reviewer for the comments.
Mitochondria have been implicated in the process of carcinogenesis, which includes alterations of cellular metabolism and cell death pathways. Alterations of mitochondrial networks are involved directly or indirectly in processes resulting in hypoxia-tolerant and hypoxia-sensitive gliomas, and by the hypoxia-inducible factor-1 (HIF-1), glycolytic protein isoforms, and fatty acid synthase (Arismendi-Morillo, 2009). In addition, the mitochondria in cancer cells have been observed with lucent-swelling matrix associated with disarrangement and distortion of cristae and partial or total cristolysis (Arismendi-Morillo, 2009; Arismendi-Morillo, 2011), supporting the presence of damaged mitochondria in cancers (Ricci et al., 2021; Signorile et al., 2019). Mitochondrial changes are associated with mitochondrial-DNA mutations, tumoral microenvironment conditions and mitochondrial fusion-fission disequilibrium (Arismendi-Morillo, 2009). In colorectal cancer cell lines, knockdown of MCT2 causes mitochondrial dysfunction, cell-cycle arrest, and senescence without additional cellular stress (Ricci et al., 2021).
Arismendi-Morillo G. Electron microscopy morphology of the mitochondrial network in human cancer. Int J Biochem Cell Biol. 2009 Oct;41(10):2062-8. doi: 10.1016/j.biocel.2009.02.002. Epub 2009 Feb 13.PMID: 19703662
Arismendi-Morillo G. Electron microscopy morphology of the mitochondrial network in gliomas and their vascular microenvironment. Biochim Biophys Acta. 2011 Jun;1807(6):602-8. doi: 10.1016/j.bbabio.2010.11.001.PMID: 21692239.
Ricci F, Corbelli A, Affatato R, Chilà R, Chiappa M, Brunelli L, Fruscio R, Pastorelli R, Fiordaliso F, Damia G. Mitochondrial structural alterations in ovarian cancer patient-derived xenografts resistant to cisplatin. Am J Cancer Res. 2021 May 15;11(5):2303-2311. eCollection 2021.PMID: 34094686
Signorile A, De Rasmo D, Cormio A, Musicco C, Rossi R, Fortarezza F, Palese LL, Loizzi V, Resta L, Scillitani G, Cicinelli E, Simonetti F, Ferretta A, Russo S, Tufaro A and Cormio G. Human ovarian cancer tissue exhibits increase of mitochondrial biogenesis and cristae remodeling. Cancers. 2019; 11: 1350